# An Experimental Study on Antioxidant Enzyme Gene Expression in *Trematomus newnesi* (*Boulenger, 1902*) Experimentally Exposed to Perfluoro-Octanoic Acid

**DOI:** 10.3390/antiox12020352

**Published:** 2023-02-02

**Authors:** Sara Pacchini, Elisabetta Piva, Sophia Schumann, Paola Irato, Daniela Pellegrino, Gianfranco Santovito

**Affiliations:** 1Department of Biology, University of Padova, 35131 Padova, Italy; 2Department of Biology, Ecology and Earth Sciences, University of Calabria, 87036 Rende, Italy

**Keywords:** Antarctic fish, perfluoroalkyl substances, PFOA, oxidative stress, antioxidant defence system

## Abstract

Antarctica is the continent with the lowest local human impact; however, it is susceptible to pollution from external sources. Emerging pollutants such as perfluoroalkyl substances pose an increasing threat to this environment and therefore require more in-depth investigations to understand their environmental fate and biological impacts. The present study focuses on expression analysis at the transcriptional level of genes coding for four antioxidant enzymes (*sod1*, *sod2*, *gpx1*, and *gpx4*) in the liver and kidney of an Antarctic fish species, *Trematomus newnesi *(*Boulenger*, *1902*). mRNA levels were also assessed in fish exposed to 1.5 μg/L of perfluoro-octanoic acid for 10 days. The kidney showed a higher level of expression than the liver in wildlife specimens. In the liver, the treatment induced an increase in gene expression for all the considered enzymes, whereas in the kidney, it induced a general decrease. The obtained results advance the scientific community’s understanding of how the potential future presence of anthropogenic contaminants in the Southern Ocean can affect the antioxidant system of Antarctic fishes. The presence of pollutants belonging to the perfluoroalkyl substances in the Southern Ocean needs to be continuously monitored in parallel with this type of research.

## 1. Introduction

The effect of anthropogenic chemical pollution on ecosystems has attracted growing attention from the scientific community in recent years. Antarctica is regarded as an “irreplaceable natural laboratory” due to its geographic and climatic isolation, as it is the only non-anthropized continent subject to scarce local sources of pollution (due to scientific research facilities, fishing, and tourism) but relatively high loads of contaminants from other areas [1,2].

Because of their potential toxicity, per- and polyfluoroalkyl substances (PFAS), a heterogeneous group of synthetic chemicals, are of particular concern as emerging pollutants due to their worldwide presence, propensity for transport, bioaccumulation, environmental persistence, and potential toxicity. These factors are caused by their unique chemical and physical properties (especially the strong chemical stability of C-F bonds), which give them thermal and chemical stability, as well as resistance to biodegradation [3]. Perfluoro-octanoic acid (PFOA) is part of the emerging, most prevalent, and widespread PFAS (included in the 2019 list of POPs regulated by the Stockholm Convention, Annex A) [4]. It is an ionizable substance with greater solubility than chlorinated POPs, and consequently, it can reach even the most remote areas of the planet, such as Antarctica, mostly through oceanic currents (OCT) [5]. For this reason, the aquatic ecosystem is one of the major sinks of environmental PFAS because of their high water solubility in comparison to traditional POPs, although we know that several matrices, including sediments, plants, and biota, contain these contaminants [6,7]. However, to the best of our knowledge, the data on the occurrence of PFAS in Antarctic biota are very limited, so more studies are required to investigate their occurrence and trophodynamic behaviors in Antarctic ecosystems [7].

Species belonging to the suborder *Notothenioidei* were chosen as experimental models, given that this group of teleosts is the most widespread among Antarctic fish fauna, representing 91% of the biomass on the continental shelf and 45% in the Southern Ocean [8]. These endemic organisms have great ecological importance, as they developed physiological adaptations to extreme environmental conditions, such as temperatures below the freezing point of body fluids, limited amounts of nutrients, high hydrostatic pressures, high dissolved oxygen concentrations, and the constant presence of an ice-covered surface [9,10]. Nevertheless, it is not so much the severe conditions but their stability that makes the Antarctic environment an ideal study area, as organisms there have evolved with a smaller number of variables than in other environments. In this way, endemic organisms may consequently exhibit less phenotypic flexibility and, hence, be more susceptible to environmental perturbations [8]. Some of the major physiological adaptations developed by Antarctic fishes are the absence of a swim bladder, with a consequent compensation through partially cartilaginous bones and accumulation of fat deposits; the ability to biosynthesize antifreezing compound; the molecular specialization of tubulins; and the reduction in the concentration of hemoglobin and the number of erythrocytes in order to reduce the viscosity of blood [11].

Furthermore, our particular interest in the antioxidant system of Antarctic marine organisms arises from its high efficiency [12]. In fact, the low temperature and salinity of seawater can determine an elevated solubility of oxygen, with a consequent high bioavailability of this gas and a greater probability of the formation of reactive oxygen species (ROS) with higher oxidative stress risk [11]. 

It is known that all aerobic organisms have evolved metabolic strategies aimed at reducing the toxicity of reactive oxygen species (ROS), natural byproducts of aerobic metabolism and continuously produced by cells, both at the cytoplasmic and mitochondrial levels [13]. This complex defence system includes both low-molecular-weight scavengers and antioxidant enzymes (e.g., superoxide dismutase (SOD) and glutathione peroxidase, (GPx)) for the detoxification and removal of ROS [14].

Several studies have confirmed the correlation between toxicity induced by xenobiotics (such as PFAS) and an increase in ROS formation, with a consequent greater risk of oxidative stress [15]. For these reasons, variations in the content and the activity of antioxidant enzymes are useful as biomarkers for oxidative stress caused by contaminants in a variety of marine organisms [16]. PFAS can bioaccumulate and biomagnify along food webs, affecting ROS production at the molecular level and leading to an imbalance in cellular redox [7]. This may result in cell death, DNA damage, altered enzymatic function, and consequent oxidative damage to the entire organism [17,18].

In the current study, we examined the mRNA accumulation of some SOD and GPx isoforms in *Trematomus newnesi* with the goal of assessing the gene expression of these antioxidant defence system components induced by PFOA at the transcriptional level. Different isoforms were chosen to highlight possible differences in enzyme expression at the subcellular level.

The considered organs are the liver and kidney due to their physiological role in the accumulation and excretion of xenobiotics, respectively [3].

The obtained results demonstrate, for the first time in Antarctic fish experimentally exposed to PFAS, how this substance is able to activate antioxidant responses consistent with the possible increase in ROS formation in PFAS-targeted cellular compartments, such as the mitochondria. Previous research in mammals verified that more than 90% of ROS are generated in mitochondria, and approximately 2% of the whole O_2_ uptake is accounted for by ROS generated as obligatory byproducts of oxidative metabolism. Mitochondria are the most important subcellular sites for primary superoxide anion (^•^O_2_^−^) generation, with subsequent production of hydrogen peroxide (H_2_O_2_) by the catalytic action of the mitochondrial SOD2 and, eventually, also of significant amounts of the extremely toxic hydroxyl radical (HO•) [19].

Furthermore, of the two organs considered, the most evident results involved the liver, as expected, from a response to acute stress.

## 2. Materials and Methods

### 2.1. Ethical Procedures

The sample collection and animal research performed in this work are in accordance with the Antarctic Treaty Protocol on Environmental Protection, Annex II, Art. 3, and with the Italian Ministry of Education, University, and Research regulations regarding activities and environmental protection in Antarctica. All experiments were carried out in accordance with the U.K. Animals (Scientific Procedures) Act (1986) and its related recommendations, EU Directive 2010/63/EU, and Italian DL 2014/26 for animal experiments.

### 2.2. Sampling Activity 

The current research is based on fieldwork that was performed during the XXXVII Italian Antarctic Expedition at Mario Zucchelli Station in Terra Nova Bay, Antarctica (74°42′ S, 167°7′ E). Adult specimens of *T. newnesi* were caught at Tethys Bay (74°42,001′ S, 164°02, 640′ E) at a depth of approximately 62 m. The collected specimens were between 20 and 23.5 cm in length and between 83.8 and 149.7 g in weight. Casually, all the sampled organisms were females. The taxonomic identification of the specific target species was performed using morphological features described in two atlases by Gon and Heemstra [20] and by Miller [21]. In particular, we considered the number of scales in various parts of the body and the number of rays in the fins.

After the sampling, the specimens were held in thermostated aquariums filled with oxygenated seawater at a temperature of approximately −1 °C. After a distressing period of four days, two experimental groups were established: a control group consisting of ten untreated organisms and an experimental group of ten organisms exposed to 1.5 μg/L of PFOA (Sigma-Aldrich, Auckland, New Zealand; initially in powder form and later diluted in seawater: stock solution in a ratio of 100 mg/100 mL) for ten days. The PFOA concentration was the same as that used in a previous work with other Antarctic organisms, such as bivalve mollusks and macroalgae. It is a very high concentration with respect to that found in the Antarctic environment, but the aim of the experiment was to induce sufficient PFOA accumulation in short time. In fact, the time of exposure was only 10 days because logistic support permitted this maximum time. However, it is of note that this concentration did not cause high toxicity and mortality.

After this exposure time, the organisms were sacrificed with an overdose of ethyl 3-aminobenzoate methanesulfonate salt (0.065 g/L; Sigma; St. Louis, MO, United States), then dissected for the collection of the liver and kidney, which were then kept at −80 °C after being frozen in liquid nitrogen.

### 2.3. Primer Design; Total RNA Extraction; and sod1, sod2, gpx1, and gpx4 cDNA Synthesis

The GenBank database on the NCBI website was used to obtain the nucleotide sequences for *sod1*, *sod2*, *gpx1*, and *gpx4*, as well as *gapdh* for *T. newnesi.* Primers were designed in the coding regions using Primer3 software (https://primer3.ut.ee/ (accessed on 1 June 2022)). The primer pairs were then examined using an IDT Oligo Analyzer device (https://eu.idtdna.com/pages/tools/oligoanalyzer (accessed on 1 June 2022)). Starting from the tissues mentioned in the previous paragraph, the total RNA was isolated using PRImeZOL™ reagent (Canvax, Córdoba, Spain) following the manufacturer’s instructions. When working with Antarctic fish samples, a further purification step with 8M LiCl is required to eliminate the possible presence of glucose contaminants. A NanoDrop ND-1000 spectrophotometer (Thermo Fisher Scientific, Waltham, MA, USA) was used to quantify RNA concentration; to evaluate RNA integrity, an aliquot of RNA (1000 ng/μL) was run on a denaturing stained gel. Starting from 1 μg of total RNA in a 20 μL reaction mixture, cDNA was synthesized using a Biotechrabbit™ cDNA synthesis kit (Berlin, Germany). The mixture contains 2 μL of dNTP mix (10 mM), 0.5 μL of RNase inhibitor, 40 U/μL, 0.5 μL of oligodT anchor primer, 4 μL of 5x reverse transcriptase buffer, 1 μL of RNA template, 1 μL of RevertUPTM II reverse transcriptase, and pure water. The mix was then incubated at 50 °C for 1 h + 99 °C for 5 min.

We performed PCR using 50 ng of the obtained cDNA and various forward and reverse primers listed in Appendix A, as well as GRS Taq DNA polymerase (Grisp, Porto, Portugal).

The PCR was designed with under following conditions: 95 °C for 5 min, 38 cycles of 95° for 30 s, 60° for 30 s, and 72 °C for 30 s. The last step was a cycle of 72 °C for 5 min to allow for the elongation of the filaments.

### 2.4. qRT-PCR Analysis 

For each species and sample, real-time PCR was conducted to analyze the concentrations of cytoplasmic *sod1* and *gpx1* mRNAs, as well as those of mitochondrial *sod2* and *gpx4* mRNAs. For this investigation, the housekeeping gene used was *gapdh* (GenBank accession number: KF915299.1). Each cDNA sample was analyzed with the exact primers listed in Appendix A with the following program: 95 °C for 2 min, 40 × (95 °C for 20 s and 60 °C for 60 s). The last step was a dissociation stage at 95 °C for 15 s, followed by 60 °C for 1 min, 95° for 15 s, and 60 °C for 15 s.

Transcript levels were expressed (in arbitrary units (a.u.)), representing the ratio between each gene and *gapdh* expression in the same sample.

### 2.5. Statistical Analysis 

The results were derived from five biological replicates corresponding to five specimens analyzed for each experimental group. The data are reported as the mean ± standard deviation (SD). A statistical analysis of the data was performed using the Primer.exe statistical program (version 1.0, Stanton A. Glantz, McGraw Hill Education, Milan, Italy). In order to evaluate statistically significant differences between the means and standard deviations, one-way analysis of variance (ANOVA) was applied, followed by the Student–Newman–Keuls test. A *p*-value < 0.05 was considered statistically significant.

## 3. Results

For each antioxidant enzyme, the kidney of the control specimens always shows a greater level of expression (*p* < 0.05) than the respective liver. In particular, the renal mRNA levels for the SOD1 enzyme are about three times higher than the liver (Figure 1), whereas those for the SOD2 enzyme are about seventeen times higher (Figure 2), the GPX1 enzyme is about seven times higher (Figure 3), and the GPX4 enzyme is about two times higher (Figure 4).

Except for *sod1* (Figure 1), the mRNA levels of which seem to be unaffected by exposure to PFOA, all the enzymes considered experienced a statistically significant increase (*p* < 0.05) in gene expression in the liver after the treatment. In particular, PFOA-exposed specimens show *sod2* mRNA levels that are about 12 times greater in the liver than in the kidney (Figure 2), *gpx1* levels that are about 5 times higher (Figure 3), and *gpx4* levels that are about 4 times higher (Figure 4).

In the kidney, except for *gpx1* (Figure 3), mRNA levels of which remained unaltered following the treatment, exposure to PFOA causes a generally statistically significant reduction (*p* < 0.05) in mRNA accumulation. In particular, PFOA-exposed specimens have *sod1* enzyme mRNA levels that are about 60% lower compared to those in the liver (Figure 1), *sod2* enzyme mRNA levels that are about 34% lower (Figure 2), and *gpx4* enzyme mRNA levels that are about 63% lower (Figure 4).

## 4. Discussion

The Antarctic ecosystem, due to its isolation and distance from other continents, has special characteristics. Although pollutants are transported over great distances by the atmosphere to Antarctica (LRAT: long-range atmospheric transport), it is still considered a pristine environment [1].

Animals on this continent have been exposed to peculiar chemical and physical conditions, for example, low temperatures and high oxygen presence in marine waters [22]. These conditions have probably affected the adaptive strategies of these organisms during the last 10–12 million years and have, in particular, led to the evolution of very efficient cellular defence systems against oxidative stress [23,24,25].

Given that they evolved without a substantial selective pressure represented by the presence of these chemical elements, one of the key scientific problems associated with the ecophysiology of Antarctic animals is whether they have developed capacities for acclimatization towards a range of environmental concentrations of pollutants. In particular, in this work, we wanted to determine whether and how short-term exposure to high PFOA concentrations could affect the way antioxidant enzyme genes are expressed.

PFAS are suspected to be carcinogenic, and one possible mechanism of action is that they cause oxidative stress [17]. According to previous studies, detected variations in the antioxidant enzyme activities suggest that PFAS exposure might upset the delicate balance of the antioxidant system, increasing the production of ROS, which affects the mitochondria and triggers a cascade of events that amplifies cell apoptosis [3,26,27].

According to these studies, our findings demonstrate that the genes of almost all considered antioxidant enzymes (except SOD1, which shows no statistically significant difference) were transcriptionally activated in the liver following 10-day exposure to 1.5 µg/L of PFOA. The detoxifying role of the liver is enhanced if organisms are exposed to a contaminant such as PFOA. This was expected, as it is known that it is an organ that exhibits high metabolic activity, as it is crucial to the preservation of cellular homeostasis [28]. The liver’s rapid metabolism results in the generation of ROS, the abundance of which is monitored by its extraordinary capacity to express antioxidants [29]. However, it should be highlighted that the prominent ROS production that distinguishes the liver of Antarctic teleosts is probably more related to the detoxifying function of this organ than to its digestive activity, which is relatively limited and occasional due to the sporadic feeding of these fish. In fact, the liver is known to bioaccumulate xenobiotics, particularly metal ions such as Cd and Cu, which are present in high concentrations in the Antarctic seabed as a natural condition [30,31]. According to recent research, the liver is an organ that accumulates PFOA in high concentrations in fishes living in PFAS-polluted environments [18].

In the liver, GPx4 is the enzyme that exhibits the greatest increase in gene expression after exposure to PFOA, followed by GPx1 and SOD2, whereas SOD1 is unaffected. These results suggest an increase in ROS production within the mitochondria, given the very particular cellular localization of GPx4 and SOD2 [32], and confirm that mitochondria are among the main targets of PFAS toxicity [17,26,27] in fish [18]. *Sod1* is the most abundantly expressed antioxidant gene in untreated fish, and this result may be related to a high level of ^•^O_2_^−^ production at the cytoplasmic level under normal conditions [33]. The decrease in the expression of this gene during PFOA exposure in favour of an increase in the expression of *gpx1* could instead indicate that under such circumstances, there is a reduction in the rate of ^•^O_2_^−^ formation and an increase in that of H_2_O_2_.

In contrast, the kidney is not affected by PFOA exposure, and treated samples even show a tendency to have lower levels of mRNA accumulation coding for all antioxidant enzymes (except for GPx1, the expression of which did not change). Although the kidney also plays a detoxifying role, this result is most likely due to the non-involvement of this organ during the short (10-day) exposure period.

Nevertheless, a non-variation in kidney mRNA levels could be expected, although our results show a decrease. SOD1 is the enzyme in the kidney that exhibits the greatest reduction in gene expression after exposure to PFOA, followed by GPx4 and SOD2, whereas GPx1 is unaffected. There are two primary reasons why cells reduce messenger production for a certain protein. The first occurs when the organism does not require the protein. The second occurs when the cell needs to save energy that may be used at other locations in the body—the liver in this case, where it is necessary to mitigate a high oxidative risk—through the biosynthesis of SOD and GPx. Both factors probably influence the outcome. Given its role in xenobiotic excretion and the possible PFAS-induced toxicity, the kidney is a target organ for chemical substances. Because the exposure to PFOA in our experiment was acute rather than chronic, we can expect to observe a renal response following treatment that lasts more than 10 days.

## 5. Conclusions

The antioxidant enzymes considered in this study had already been characterized in various Antarctic fish species such as *T. bernacchii* (*Boulenger*, *1902*) [34,35] and *T. eulepidotus* (*Regan*, *1914*) from a structural and functional point of view [36]. The only thing that was known about *T. newnesi* physiology and that is shares with other species of Antarctic fish was that they possess antifreezing proteins within their body fluids [37].

To the best of our knowledge, the gene expression results provided here are the first on *T. newnesi* and among the first considering mRNA accumulation in Antarctic fish, although numerous studies have focused on the evaluation of antioxidant enzyme activity in Antarctic fish.

The collected data on gene expression demonstrate how the examined species can express molecular defences against excessive ROS production, which are more pronounced under natural conditions in the Antarctic marine environment.

The reported findings also provide insight into the physiological responses in Antarctic fish to PFAS exposure, specifically PFOA, and on the tissues that are the most crucial for the detoxification of these chemicals. SOD and GPx within the antioxidant defence system play a crucial role in the scavenging of superoxide radical (^•^O_2_^−^) and hydrogen peroxide (H_2_O_2_), respectively, under normal and stress conditions; however, this is just one part of the comprehensive system of defence against ROS. Therefore, in the future, it will be necessary to assess the function that additional antioxidant components, both enzymatic and non-enzymatic, especially those acting at the mitochondrial level [38], play in order to have a complete picture.

The information provided in this work can contribute significantly to the prediction of the physiological responses of these organisms to environmental changes, which can improve the general understanding of the molecular and functional evolution of Antarctic fishes. Furthermore, our analysis of gene expression may provide a basis for the use of antioxidant enzymes as indicators for both oxidative stress and PFOA exposure.

Regarding the prospects for the future, it is clear that the cellular analysis carried out in this work must be combined with biochemical assays to correlate the physiological response of the cell with measurements of cellular processes, such as enzymatic activity or the amount of ROS present in the tissue. Because it is known from research on other species that organisms surviving in naturally unfavorable conditions but not under acute stress have evolved post-transcriptional control of the gene expression of antioxidant enzymes, quantifying enzymatic activity would be certainly informative [39]. In fact, it can be inferred that a portion of the transcript is not immediately translated when significant levels of mRNA are present without a comparable presence of active protein. According to this theory, the transcript is kept in intracellular compartments such as P bodies or stress granules, where the messengers may go through degradation or future translation, respectively [39,40]. This circumstance enables the tissues to react to acute stress extraordinarily quickly, in this case by producing more of a particular antioxidant enzyme in response to an abrupt increase in the rate of the associated ROS generation. This theory has been strengthened by recent research on the stress granule nucleation proteins in marine animals [41], including Antarctic fish [42], which shows that this situation occurs particularly in organs such as the liver and muscles, which can experience stress, even in a short amount of time and, as a result, require a quick defensive action made explicit by antioxidants.

In the future, it will also be appropriate to increase research on the cellular production of ROS and the detoxifying function of antioxidant enzymes in physiological and non-physiological settings, such as exposure to environmental pollutants of anthropogenic origin capable of carrying out a pro-oxidant action, such as PFAS.

Furthermore, it will certainly be useful to expand the types of biomarkers tested, such as different antioxidant enzymes (besides SOD and GPx) in other types of organs or tissues. There is also a need to supplement cellular analysis with biochemical assays in order to correlate the physiological response of the cell with quantification of cellular processes, e.g., measurement of ROS content or active protein in the tissue. The detection of changes in biochemical responses provide a useful tool for the early assessment of environmental disturbance, which could subsequently be correlated with the amounts of pollutants present in Antarctica in order to assess the suitability of the physiological parameter of the specific organism as a biomarker of environmental stress.

## Figures and Tables

**Figure 1 antioxidants-12-00352-f001:**
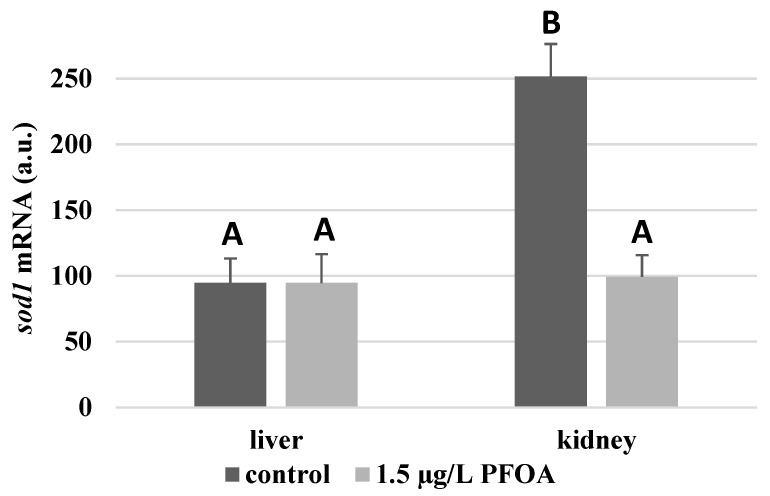
*sod1* mRNA expression levels in the liver and kidney of *T. newnesi* exposed to PFOA and unexposed (the control group). Values (arbitrary units) are indicated as mean ± SD. Transcription levels were normalized to the gapdh housekeeping gene. Different letters correspond to significant statistical differences (*p* < 0.05) among different means (Student–Newman–Keuls test), with ten specimens per experimental group.

**Figure 2 antioxidants-12-00352-f002:**
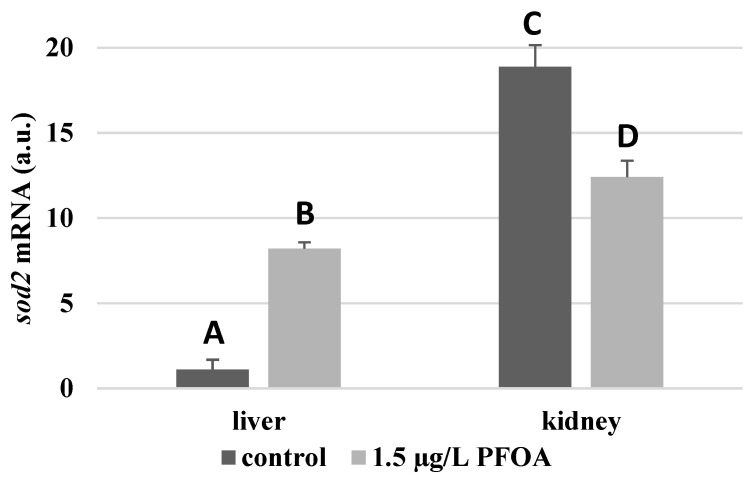
*sod2* mRNA expression levels in the liver and kidney of *T. newnesi* exposed to PFOA and unexposed (the control group). Values (arbitrary units) are indicated as mean ± SD. Transcription levels were normalized to the gapdh housekeeping gene. Different letters correspond to significant statistical differences (*p* < 0.05) among different means (Student–Newman–Keuls test), with ten specimens per experimental group.

**Figure 3 antioxidants-12-00352-f003:**
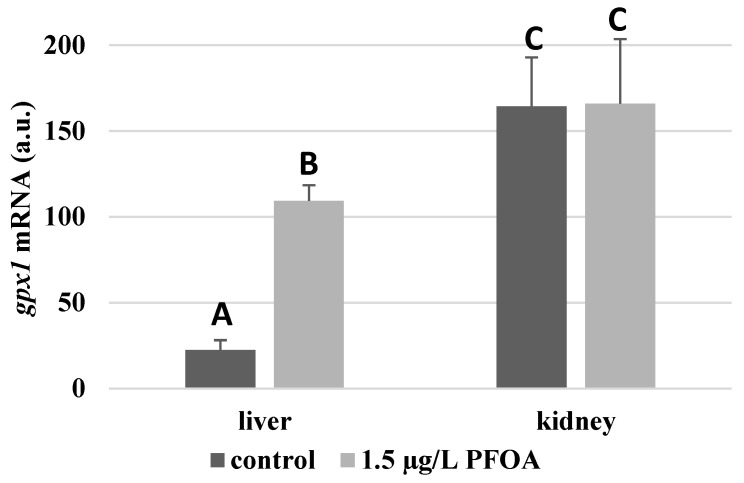
*gpx1* mRNA expression levels in the liver and kidney of *T. newnesi* exposed to PFOA and unexposed (the control group). Values (arbitrary units) are indicated as mean ± SD. Transcription levels were normalized to the gapdh housekeeping gene. Different letters correspond to significant statistical differences (*p* < 0.05) among different means (Student–Newman–Keuls test), with ten specimens per experimental group.

**Figure 4 antioxidants-12-00352-f004:**
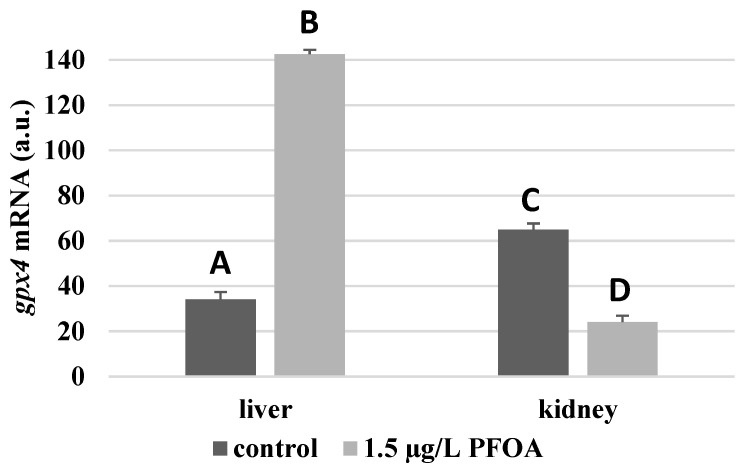
*gpx4* mRNA expression levels in the liver and kidney of *T. newnesi* exposed to PFOA and unexposed (the control group). Values (arbitrary units) are indicated as mean ± SD. Transcription levels were normalized to the gapdh housekeeping gene. Different letters correspond to significant statistical differences (*p* < 0.05) among different means (Student–Newman–Keuls test), with ten specimens per experimental group.

## Data Availability

The data are contained within the article and Appendix A.

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
