# Peer review of "An Experimental Study on Antioxidant Enzyme Gene Expression in Trematomus newnesi (Boulenger, 1902) Experimentally Exposed to Perfluoro-Octanoic Acid"

_antioxidants, 2023, doi:10.3390/antiox12020352_

Round 1

Reviewer 1 Report

The manuscript by Pacchini et al. deals with an interesting topic. It is generally well written and results are reported clearly.

Please fine some comments here below, most attention on 1 and 

1-The title attracts the reader's interest but the initial question does not find any answer. I agree that it is not easy, but at least some comments are essential if you want to keep this title. To give a contribution in this sense, the results are few and concern "only" the expression of transcripts. Therefore, literature support is needed. Otherwise, I recommend changing the title.

2-Abstract: first sentence use of “yet” and also “still” is inappropriate.

3-Line 101: “immediately” is probably a typo.

4-FIGURE 1: the caption reports sod2 mRNA, but Fig.1 regards sod1

5-Line 211: In this work (not in this paper) we aimed to verify….

6-Lines 258-259: rewrite, not clear. Although….therefore??

7-Lines 318: funding and not fund-ing.

Author Response

The manuscript by Pacchini et al. deals with an interesting topic. It is generally well-written, and results are reported clearly.

Please find some comments here below, most attention on 1 and 

1-The title attracts the reader's interest, but the initial question does not find any answer. I agree that it is not easy, but at least some comments are essential if you want to keep this title. To give a contribution in this sense, the results are few and concern "only" the expression of transcripts. Therefore, literature support is needed. Otherwise, I recommend changing the title.

We agree and, also following the suggestions of other Reviewers, the title was changed as follows: “An experimental study on antioxidant enzymes gene expression in Trematomus newnesi, experimentally exposed to perfluorooctanoic acid”.

2-Abstract: first sentence use of “yet” and “still” is inappropriate.

Correction made, Line 13: Sentence changed as follow: “Antarctica is the continent with the lowest local human impact; however, it is susceptible to pollution from external sources”.

3-Line 101: “immediately” is probably a typo.

Correction made, Line 120.

4-FIGURE 1: the caption reports sod2 mRNA, but Fig.1 regards sod1

Correction made. FIGURE 1 caption sod2 was changed with sod1.

5-Line 211: In this work (not in this paper) we aimed to verify….

Correction made, Line 241: word “paper” was changed in “work”.

6-Lines 258-259: rewrite, not clear. Although….therefore??

he sentence has been changed: Line 287: ”Since the exposure to PFOA in our experiment was acute rather than chronic, we can suppose to observe a renal response performing a treatment that lasts more than 10 days.”

7-Lines 318: funding and not fund-ing.

Correction made, Line 355.

Reviewer 2 Report

The manuscript entitled “Are Antarctic fish adapted to face global changes? A study on antioxidant enzymes gene expression in Trematomus newness from Ross Sea, experimentally exposed to PFOA” evaluate the gene expression at the transcriptional level of various isoforms of SOD (1 and 2) and GPx (1 and 4), in Trematomus newnesi, the first time in an Antarctic fish species to evaluate the expression of antioxidant enzymes in the Antartic marine environment. 

It’s a well-structured manuscript, complete, easy to follow and understand. Is a very interesting and innovative topic. In my opinion, some minor correction should be applied in the text before publication.

 Below some minor corrections are reported.

Abstract

Line 17-18 Please add the two different organs investigated in the study

Introduction

Line 38 Add a full name of “PFOA” as the main substance

Line 75-85 most of the components are redundant in this part, because they are already included in the materials and methods section, instead it is important to deal with the purpose of the work by including what is expected throughout the text, as done in the abstract

Materials and methods

Line 104 “Their specimens were 20-23.5 cm in length and 83.8-149.7 g” are this referred to the experimental group or both?

Line 114-116 Delete one bracket

Line 144-146 This section should be described in a more specific manner

Results and Discussion

Line 205-206 All the references should be placed at the end of the sentence

Line 220-243 Please respect always the same format in writing some parameters “SOD1; GPx4…) throughout the text

Line 262-266 Why was this little-studied and little-known species chosen precisely to assess globe changes? It’s known that  to assess the adaptation status of an animal, it must take into account previous studies on the same subject evaluating the variations over time.

Author Response

The manuscript entitled “Are Antarctic fish adapted to face global changes? A study on antioxidant enzymes gene expression in Trematomus newnesi from Ross Sea, experimentally exposed to PFOA” evaluate the gene expression at the transcriptional level of various isoforms of SOD (1 and 2) and GPx (1 and 4), in Trematomus newnesi, the first time in an Antarctic fish species to evaluate the expression of antioxidant enzymes in the Antarctic marine environment. 

It’s a well-structured manuscript, complete, easy to follow and understand. Is a very interesting and innovative topic. In my opinion, some minor corrections should be applied in the text before publication.

Below some minor corrections are reported.

Abstract

1) Line 17-18 Please add the two different organs investigated in the study

Correction made. The sentence has been changed as follows: Lines 16-18 “The present study focuses on expression analysis at the transcriptional level of genes coding for 4 antioxidant enzymes (sod1, sod2, gpx1, gpx4) in the liver and kidney of an Antarctic fish species, Trematomus newnesi (Boulenger, 1902)”.

Introduction

2) Line 38 Add a full name of “PFOA” as the main substance

Correction made, Line 41: PFOA was changed to “Perfluorooctanoic acid”.

3) Line 75-85 most of the components are redundant in this part, because they are already included in the materials and methods section, instead it is important to deal with the purpose of the work by including what is expected throughout the text, as done in the abstract

Correction made. The paragraph indicated was changed as follows: Lines 86-97 “In the current study, we examined the mRNA accumulation of some SOD and GPx isoforms in Trematomus newnesi with the goal of assessing the gene expression, at the transcriptional level, induced by PFOA, of these antioxidant defence system components. The different isoforms were chosen to highlight possible differences in enzyme expression also at the sub-cellular level.

The considered organs were the liver and kidney, due to their physiological role respectively in the accumulation and excretion of xenobiotics [3].

The obtained results allowed us to demonstrate, for the first time in Antarctic fish experimentally exposed to PFAS, how this substance can activate antioxidant responses consistent with the possible increase in ROS formation in PFAS-targeted cellular compartments, such as the mitochondria. Furthermore, of the two organs considered, the most evident results involved the liver, as expected from a response to acute stress.”

Materials and methods

4) Line 104 “Their specimens were 20-23.5 cm in length and 83.8-149.7 g” are this referred to the experimental group or both?

Correction made, line rewritten for clarity, Lines 112-113 “The collected specimens were between 20-23.5 cm in length and 83.8-149.7 g in weight.”

5) Line 114-116 Delete one bracket

Correction made, Line 141.

6) Line 144-146 This section should be described in a more specific manner

Correction made, Lines 170-176: “The results derive from five biological replicates, corresponding to five specimens analyzed for each experimental group. The data are reported as the mean ± standard deviation (SD). A statistical analysis of the data was performed using the Primer.exe statistical programme (version 1.0, Stanton A. Glantz, McGraw Hill Education, Milan, Italy). In order to evaluate statistically significant differences between the means and standard deviations, one-way analysis of variance (ANOVA), was applied, followed by the Student-Newman-Keuls test. A p-value < 0.05 is considered statistically significant.”

Results and Discussion

7) Line 205-206 All the references should be placed at the end of the sentence

Correction made, Line 236.

8) Line 220-243 Please respect always the same format in writing some parameters “SOD1; GPx4…) throughout the text

It is well known that the way to write proteins and genes names are different. When we referred to the proteins, we used capital letters (SOD1; GPx4…). When we refered to genes, we used lowercase and italic letters. Therefore, we maintained the same format already present in the text.

9) Line 262-266 Why was this little-studied and little-known species were chosen precisely to assess global changes? It’s known that to assess the adaptation status of an animal, it must take into account previous studies on the same subject evaluating the variations over time.

The aim of our experimental research was to study the antioxidant defence components of T. newnesi and how this system can be influenced on a biomolecular level by exposure to certain chemical stressors. Unfortunately, knowledge of the effect of perfluoroalkyl substances on Antarctic fauna is lacking.

The target species of our work is one of the components of the Antarctic fauna, and it is our opinion that, in order to properly assess the effects of global changes on an ecosystem, it's necessary to consider each element of it. Therefore, we believe that our work is an important contribution to increasing our knowledge of the Antarctic ecosystem and the effects these substances have on it.

It is certainly not possible to deduce the effect of global changes solely by considering T. newnesi, which is why we have also changed the title in response to specific requests from other Reviewers. The new title was changed as follows: "An experimental study on antioxidant enzymes gene expression in Trematomus newnesi, experimentally exposed to perfluorooctanoic acid “.

Reviewer 3 Report

Article:

Are Antarctic fish adapted to face global changes? A study on antioxidant enzymes gene expression in Trematomus newnesi from Ross Sea, experimentally exposed to PFOA

Manuscript ID:

antioxidants-2168015

General Comments

Sara Pacchini and colleagues performed a study to detect the transcriptional levels of antioxidant enzymes in kidney and liver of an Antarctic teleost after the exposure to PFOA. The paper focuses an interesting aspect of the antioxidant response of this antarctic teleost. Despite this, it is my opinion that the paper should be considered as an experimental study of antioxidant enzymes response, more than a study on the ability of Antarctic fish to face global changes. Moreover, I suggest the authors to use more caution in proposing biomarkers in future Antarctic biomonitoring programs. This, especially because the antioxidant responses in natural environments can be ascribed to different stressors, and then the variation in the antioxidant enzymes transcriptional levels could be related to other factors rather than PFOA exposure. This is particularly true considering what the authors assess in the introduction (lines 44-46): “However, to our knowledge, the data on the occurrence of PFAS in Antarctic biota are very limited, so more studies are required to investigate their occurrence and trophodynamic behaviours in Antarctic ecosystems”.

Specific comments

Title:

The title should be changed excluding the opening question “Are Antarctic fish adapted to face global changes?”. Also, please avoid the use of acronyms in the title.

Abstract:

Please avoid the use of acronyms in the abstract.

Lines 19-20: maybe this sentence should be moved before (end of the line 18).

Consider changing abstract according to previous “general comments”

Keywords:

Keywords should not contain terms already used in the title. Please provide different keywords

Introduction:

The authors well introduce the focused topic. Despite this, it is my opinion, that the introductory part regarding the AntaGPS project should be deleted, it is not relevant to the study.

Moreover, lines 75-85 are superfluous. It is my suggestion to conclude the introduction with the research question of the study regarding the response of antioxidant enzymes transcriptional levels after the exposure to PFOA.

Material and methods:

Material and methods section is well organized. However, it is not possible to find the taxonomical identification method used and the reason for selecting the PFOA concentration and time exposure (evidence of such PFOA environmental concentrations or other reasons?).

No information are present about the PFOA (composition, provider, method of dilution etc.).

Line 105: why all the fish were female? Is there a way to identify the specimen sex before dissecting them? If yes, in the case of evident morphological sexual dimorphism, please provide a description and relative references.

Results:

The results are clearly presented.

Discussion:

Results are very well discussed, with coherent and updated bibliography.

Other minor comments

It is my suggestion to use the complete zoological nomenclature each time a species is mentioned for the first time, e.g. Trematomus newnesi (Boulenger, 1902).

Author Response

General Comments

Sara Pacchini and colleagues performed a study to detect the transcriptional levels of antioxidant enzymes in kidney and liver of an Antarctic teleost after the exposure to PFOA. The paper focuses an interesting aspect of the antioxidant response of this antarctic teleost. Despite this, it is my opinion that the paper should be considered as an experimental study of antioxidant enzymes response, more than a study on the ability of Antarctic fish to face global changes. Moreover, I suggest the authors to use more caution in proposing biomarkers in future Antarctic biomonitoring programs. This, especially because the antioxidant responses in natural environments can be ascribed to different stressors, and then the variation in the antioxidant enzymes transcriptional levels could be related to other factors rather than PFOA exposure. This is particularly true considering what the authors assess in the introduction (lines 44-46): “However, to our knowledge, the data on the occurrence of PFAS in Antarctic biota are very limited, so more studies are required to investigate their occurrence and trophodynamic behaviours in Antarctic ecosystems”.

We agree with this comment and have therefore changed the text to focus only on the physiological aspect without referring to biomarkers of exposure to environmental contaminants and biomonitoring programs.

Specific comments

Title:

1) The title should be changed excluding the opening question “Are Antarctic fish adapted to face global changes?”. Also, please avoid the use of acronyms in the title.

à The title was changed as follows: “An experimental study on antioxidant enzymes gene expression in Trematomus newnesi, experimentally exposed to perfluorooctanoic acid”.

Abstract:

2) Please avoid the use of acronyms in the abstract.

Correction made. We eliminated the PFAS and PFOA acronyms. We maintained sod1, sod2, gpx1, gpx4 because these are the names of the considered genes.

3) Lines 19-20: maybe this sentence should be moved before (end of the line 18).

Correction made, Lines 18-19.

4) Consider changing the abstract according to previous “general comments” ­

Correction made. We changed the abstract and eliminated the following sentence: “[…] for using the expression of antioxidant enzymes as biomarkers, both of oxidative stress and exposure to PFAS, in future biomonitoring campaigns in the Antarctic marine environment.”.

The abstract is now changed as follows: Antarctica is the continent with the lowest local human impact; however, it is susceptible to pollution from external sources. Emerging pollutants, like perfluoroalkyl substances, pose an increasing threat to this environment and therefore require more in-depth investigations to understand their environmental fate and biological impacts. The present study focuses on expression analysis at the transcriptional level of genes coding for 4 antioxidant enzymes (sod1, sod2, gpx1, gpx4) in the liver and kidney of an Antarctic fish species, Trematomus newnesi (Boulenger, 1902). The mRNA levels were also assessed in fish exposed to 1.5 μg/L of perfluorooctanoic acid for 10 days. The kidney showed a higher level of expression than the liver in wildlife specimens. In the liver, the treatment induced an increase in gene expression for all the considered enzymes, while in the kidney it induced a general decrease. The obtained results advance the scientific community’s understanding of how the potential future presence of anthropogenic contaminants in the Southern Ocean can affect the antioxidant system of Antarctic fish. The presence of pollutants, belonging to the perfluoroalkyl substances, in the Southern Ocean will certainly need to be continuously monitored in parallel with this type of research.”

Keywords:

5) Keywords should not contain terms already used in the title. Please provide different keywords

Correction made. We changed the keywords as requested.

Introduction:

6) The authors well introduce the focused topic. Despite this, it is my opinion, that the introductory part regarding the AntaGPS project should be deleted, it is not relevant to the study.

Correction made. We deleted the following part: “This research is part of the "AntaGPS" project, funded by the PNRA (National Research Program in Antarctica), which uses Antarctica as a global pollution sensor and its endemic organisms as bioindicators. One of the objectives of the project is the study of the antioxidant defence components of Antarctic fish and how this system can be influenced, at the biomolecular level, by exposure to some chemical stress factors”.

7) Moreover, lines 75-85 are superfluous. It is my suggestion to conclude the introduction with the research question of the study regarding the response of antioxidant enzymes transcriptional levels after the exposure to PFOA.

Correction made. We changed this part also according to the comments of Reviewer number 1.

Material and methods:

8) The material and methods section is well organized. However, 1) it is not possible to find the taxonomical identification method used and 2) the reason for selecting the PFOA concentration and time exposure (evidence of such PFOA environmental concentrations or other reasons?).

1) This part was added to the text. Lines 114-118 ”The taxonomic identification of the specific target species was performed using two atlases: Fishes of the Southern Ocean. Gon O. and Heemstra P. C. Grahamstown, South Africa, J.L.B. Smith Institute of Ichthyology, (1990) and History and Atlas of the Fishes of the Antarctic Ocean. Miller R. Foresta Institute for Ocean and Mountain Studies, Carson City, Nevada (1993).”

2) This part was added to the text. Lines 124-130 “The PFOA concentration was the same already used in previous work with other Antarctic organisms, such as bivalve molluscs and macroalgae. It is a very high concentration with respect to what we can find in the Antarctic environment, but the aim of the experiment was to induce enough PFOA accumulation in a short time. In fact, the time of exposure was only 10 days because logistic support permitted this maximum time. However, it is to note that this concentration did not cause high toxicity and mortality.”

9) No information is present about the PFOA (composition, provider, method of dilution etc.).

We added this information to the text. Lines 123-124 “PFOA (Sigma-Aldrich, Auckland, New Zealand; initially in powder form, it was diluted in seawater: stock solution 100 mg / 100 mL).”

10) Line 105: why all the fish were female? Is there a way to identify the specimen sex before dissecting them? If yes, in the case of evident morphological sexual dimorphism, please provide a description and relative references.

It’s not possible to identify the sex before dissecting them because there is no sexual dimorphism therefore only after the dissection the gonad morphology makes it possible to discriminate between females and males.

Results:

The results are clearly presented.

Discussion:

Results are very well discussed, with coherent and updated bibliography.

Other minor comments

11) It is my suggestion to use the complete zoological nomenclature each time a species is mentioned for the first time, e.g. Trematomus newnesi (Boulenger, 1902).

We added the complete zoological nomenclature for each mentioned species. Lines 18, 292 and 293.

Round 2

Reviewer 3 Report

Dear Authors, 

thanks for consider my comments. II would like to suggest only the following points:

- in the title add species authorities Trematomus newnesi (Boulenger, 1902)

- use of term "fishes" in the case it is referred to different fish species, as in the abstract happens (line 24). 

All the best regards

Author Response

- in the title add species authorities Trematomus newnesi (Boulenger, 1902)

We added the correction suggested. The title was changed as follows: “An experimental study on antioxidant enzymes gene expression in Trematomus newnesi (Boulenger, 1902), experimentally exposed to perfluorooctanoic acid”.

- use of the term "fishes" in the case it is referred to different fish species, as in the abstract happens (line 24). 

We used the term fishes in the Abstract (Line 24) and in lines 24, 65, 271 and 301.